# Understanding Resource Consumption in the Home, Community and Society through Behaviour and Social Practice Theories

**Jessica K. Breadsell** *[ID], **Christine Eon**[ID] and **Gregory M. Morrison**[ID]

Curtin University Sustainability Policy Institute, School of Design and the Built Environment, Curtin University, Perth 6102, Australia; christine.eon@curtin.edu.au (C.E.); greg.morrison@curtin.edu.au (G.M.M.)

* Correspondence: Jessica.breadsell@curtin.edu.au

**Abstract:** The practices and behaviours of individuals influences resource consumption at many scales and are shaped by a multitude of psychological, social, and technical factors. This conceptual paper examines the differences between socio-psychological and social practice theories, building on the Chalk and Cheese debate in the literature. Insight is provided into their potential value in understanding resource consumption studies at different scales: the individual, the home, community, and societal. Each theory has its own qualitative and quantitative methods which allude to different conclusions and recommendations for resource consumption initiatives. We review the debate surrounding the application of both theories, adding our voice to the potential for both theories to be used at different scales and for different time periods, along with comments on the interlocking nature of practices. Design and technology changes can lead to quicker changes in behaviour and practices, whereby socio-psychological theories offer insights into changes in mind frame, values, and social norms.

**Keywords:** social practice theory; behaviour change; resource consumption; scale; home; community; society

## 1. Introduction

This publication reviews socio-psychology theories and social practice theory (SPT) as well as methods commonly employed to influence occupants in their use of resources. The objective of this article is to discuss how these different theories can be successful to promote sustainable consumption at various scales. The term sustainable consumption refers to the consumption of more efficiently or ethically produced goods where consumers consider environmental and social aspects before purchase [1]. Reducing resource consumption to more sustainable levels has been identified as a vital path to address the pressing issues of climate change, over-exploitation of dwindling resources, and subsequent environmental impacts [2].

Sustainable consumption may be encouraged through changes multiple practices including housing, food, waste or mobility practices or individual actions involving energy and water consumption. The reduction of resource use in households is considered by many as a cost-effective step toward urban sustainability [3]. Improved building envelope design and technology are known to significantly reduce energy and other resource demand in households. However, they are not the only influencing factors [4,5]. Individuals are a key player in making everyday decisions and influencing usage through their behaviours and practices, which, in turn, are influenced by place, technologies, interpersonal relationships, society and information [6]. The implementation of strategies based on behaviour change has been a key approach to promoting resource efficiency in residential buildings.

Finding optimal behaviour change methods, in particular, targeting energy conservation has been the focus of research since the 1970s [7]. However, traditional socio-psychological approaches to enforcing sustainable consumption have failed to drive society-wide changes necessary to limit catastrophic environmental impacts [8,9].

The ideas behind changed behaviour for environmental sustainability in the home have generally been based on social theories [10–12]. Typically, socio-psychology approaches consider cognitive dissonance, social norms, information provision, and feedback to instigate change [13]. Since human-computer interactions became accessible, scientists have started deploying them on a regular basis for the delivery of eco-feedback and information. This can be referred to as persuasive sustainability [14]. This model has been sustained by economists and psychologists and has permeated its way into both the UK and the Australian policy context [15,16]. Human-computer interactions are now being deployed in smart technology and homes around the world in an effort to persuade residents to alter their behaviours and practices, and subsequent resource consumption. Change in policy and societal resource use is difficult as policymakers insist that any alternative approaches are translated into the language and terminology that they are already familiar with through behaviour change programs [17,18].

SPT is an alternative theory to socio-psychology and has been prominent in sustainable consumption debates recently. SPT originates in social theory [19] but offers an alternative approach to promoting a reduction in resources consumed. SPT scholars argue that resource consumption depends largely on the practice used to carry out activities [20]. A practice is a routinised action that is composed of a number of interconnected elements: meaning, skill, and technology [6,21]. In SPT, the practice is the unit of analysis, and change can be made to practice through the alteration of one or more of the elements. The innovation of a product or procedure can, therefore, act as an enabler of sustainability, especially when designed in conjunction with users. This is referred to as non-persuasive sustainability [14], although we prefer the term enabling sustainability.

Socio-psychology theories view resource consumption as something that an individual uses depending on their personal values as well as norms, which are influenced by society. SPT approaches focus on the practices and bundles of practices that resources are involved in when these practices are performed [20]. There has been a provocative debate in the literature around the merging of socio-psychology theories with SPT for use in resource consumption studies and policy contexts, termed the Chalk and Cheese Debate [15,22,23]. Each theory has its own qualitative and quantitative methods which allude to different conclusions and recommendations [24]. In this conceptual paper, we argue that rather than being antagonising theories, both theories have a place when applied within the home space. SPT provides a valuable addition along with socio-psychology theories in understanding the complex dynamics between resource consumption and human actions [25–29].

This paper starts with a narrative literature review of SPT and socio-psychology theories that consider domestic resource use and consumption. A narrative literature review is an established approach to exploring the literature surrounding a topic in both the socio-psychology and SPT domains, allowing in-depth insights to be obtained [30]. The search terms used include: Social practice theory, practice theory, socio-psychology theory, psychology theory, theory of planned behaviour, cognitive dissonance, social norms, habitual behaviour, habits in the home, and timing of routines. However, the topics have been explored mostly through starting with seminal papers and exploring the references as needed. This review is followed by a discussion of the Chalk and Cheese debate, a review of the state of the literature surrounding the influence of scale and time on the performance of behaviours and practices and finally, comments on how the two theories insights can be applied at the scale of the home, the community and society.

## 2. Social Theories

SPT and traditional socio-psychology theories are described in this section through a narrative literature review. The main point of difference between them is the unit of analysis: socio-psychology

theories focus on the individual while SPT focuses on practice as the unit of interest. This review does not comment on the political influences and ideologies of the two theories as it is outside the scope of this research.

## 2.1. Socio-Psychology Theories

Socio-psychology is the study of people's interactions with the wider society and examines how individual behaviours, thoughts and attitudes are influenced by others, either consciously or subconsciously through social and cultural norms. Several socio-psychology theories have been developed since the 1920s and are still employed today to explain individual behaviour. These theories are frequently used for the development of pro-environmental behaviour change programs and include the theories of planned behaviour, cognitive dissonance, social norms and habitual behaviour. There is often an assumed linear relationship between information/awareness, attitudes and respective behaviours [18]. It is assumed in socio-psychological literature that individual change will result in social and resource use change [31].

The theory of planned behaviour predicts that behaviour is preceded by the attitude towards the behaviour (i.e., beliefs and evaluation of the outcomes), subjective norms (i.e., the perception of the behaviour by others) and perceived behaviour control [10]. This means that an individual who is concerned about carbon emissions, for example, might not act to reduce these due to a lack of perceived personal impacts. On the other hand, the likelihood of engagement leading to behaviour change may increase if an individual is consciously supportive of the cause or if the individual simply agrees to take action [32]. This is in accordance with the theory of cognitive dissonance which posits that people are uncomfortable to find themselves in a situation in which their attitude and behaviour are inconsistent and will, therefore, make changes towards correcting this discrepancy [11]. These adjustments can be made through changing behaviour, changing beliefs or creating new cognitive elements aligned with the behaviour [11]. This need for consistency is recognized as an opportunity to encourage behaviour change through triggering individuals' values and self-concepts, effectively making them aware of potential dissonances [33]. This can be employed through the use of nudging or prompts [34].

Whilst personal values and beliefs affect people's behaviours, individual conduct is also influenced by the behaviours and judgment of the wider society. These unspoken social rules are referred to as social norms, which are of two kinds: descriptive and injunctive [12]. Descriptive norms define what the customary behaviour is in a given situation. Injunctive norms, on the other hand, prescribe how one should behave either by approving or disapproving of the behaviour. For example, a study on littering showed that people are more inclined to drop litter in littered locations. In contrast, clean environments tend to remain unlittered for longer periods of time [12]. Social norms are even more effective when encouraged by a peer in the form of a social intervention [35].

The theories of planned behaviour and cognitive dissonance consider individuals as purely rational, evaluating outcomes as well as the costs and benefits of certain decisions, however, daily habits can prevent long-lasting change [36]. Habits are prompted to meet a specific goal and if the goal is met in a satisfactory manner, the tendency is for individuals to repeat the same behaviour on the following occasion when the same goal is being sought [37]. Repetition requires less mental effort, which can lead to unintentional habits forming and once habits are established, future actions are likely to be guided by them, regardless of values, attitudes or norms. Due to the unconscious nature of habits, habitual behaviours are only reviewed when provoked (or nudged) or in the event of a change in context [34,36]. This change in context for modifying behaviours is also what SPT advocates as a way to influence resource consumption.

Most interventions from socio-psychology theories fall into three categories, referred to here as social, technological and knowledge-based interventions. Social interventions involve some form of social interaction, such as face-to-face meetings, audits or workshops. Unilateral impersonal communication such as letters, emails, bills or marketing campaigns has been categorized as knowledge-based interventions. Technological interventions are methods that rely on technology and

do not involve any kind of social interaction. That is, in-home displays (IHDs), websites and automatic messages deliver feedback, norms, prompts, nudges and goal setting. IHDs can break down energy usage by appliance and show energy consumption in different formats, catering to different audiences while providing real-time and long-term feedback. Some researchers argue that this method enables the interaction of households with the data and therefore higher engagement [38] and appliance control [39], leading to a significant reduction of electricity consumption [40,41]. However, arguments against the deployment of IHD's focus on the fact that displays are designed by researchers and do not necessarily correspond to what the user wants to see, reverting to the background situation after a novelty period [14]. Brynjarsdottir and colleagues argue that this technology is trying to persuade the user to change rather than providing solutions to change [14]. In addition, the deployment of IHDs to influence behaviour assumes that the user has previous knowledge of interaction with this technology and that the user makes this part of everyday practice. Ongoing research is occurring on how to improve IHDs. However, there are marked differences in individual use even within households so a one-size-fits-all approach seems hard to design [41–43]. There appears to be a level that residents reach in their behaviour changes influenced by the IHDs after which more change is resisted because it disrupts routinised practices in a way which is uncomfortable or unworkable [42,44].

*2.2. Social Practice Theory*

SPT was proposed in the early 2000s as an alternative to socio-psychology theories [6,19,45–53]. SPT posits that the world is populated by social practices and their interconnected elements [16]. Human behaviour is not the result of rational choice but of the many half-conscious and highly routinised actions people take in their everyday life [54].

Individuals do not use resources such as water or energy directly, but rather with the objective of achieving a desired social outcome, an everyday practice such as cleaning, shopping or dining. In order to understand domestic resource consumption, it is therefore important to comprehend the practices involved in achieving daily objectives [25]. SPT views practices as the unit of analysis, being the mundane activities that make up most of our daily lives such as cooking, cleaning, laundering, personal hygiene and keeping thermally comfortable [55]. Practices are formed by three interconnected elements: technology, skills and meaning [21,56]. Technology is the artefacts that are used in the performance of the practice, skills are the know-how or competencies necessary to execute the practice and meaning is the understanding, assumptions, values and symbolic meanings associated with the practice, including the attitudes and feelings [57]. A change in practice can be achieved by altering one or more of these elements. The three-element model has been praised for its effectiveness in examining all the elements of practice, including non-human, material elements, different forms of intellectual and embodied knowledge, and cultural differences [58]. This allows for objects to be studied at the same ontological level as individuals, without the individual being the principal unit of analysis as it is in socio-psychology theories. Practices are social in the sense that they can be shared by many different individuals, through connected elements of meanings, technology and skill even if they are in different locations or across different time spans. It is through the repetition of practices that they become embedded in everyday life [59,60]. SPT moves beyond the single behaviour analysis of socio-psychology and examines the relationships between practices and their existence across space and time [20,21,61]. Practices can interlock with others to form bundles of practice or systems of practice (SOP) that can have socially shared knowledge, meaning, skills or technology [52,62,63]. This occurs when there are multiple practices taking place at the same time or in a similar order, such as a morning routine of having a shower, making breakfast and driving to work in the same way each workday. When multiple SOPs exist in the space of the home, this can be termed a Home System of Practice (HSOP) [21,56,61,64].

Everyday practices have a dynamic nature because as technologies, infrastructures and meanings change over time, existing social practices become obsolete and new practices become embedded in new routines [65]. Modern technology may influence the way a practice is performed. For example,

capsule-based coffee machines imply that consumers buy the manufactured coffee capsules for the coffee making practice [24]. Consumption is interwoven in practices as a by-product of undertaking the practice and can be hard to change due to their habitual and interlocking nature [17,52,66]. SPT argues that change in practice should occur through three main ways: a change in an element of the practice, disconnecting a practice from its interlocking counterparts or bundles, or inserting a new practice entirely to replace the old one [67].

A change in an element of a practice could be through changing the meaning (altering the need that an individual aims to fulfil and how this relates to their perception of their lifestyle, comfort and wellbeing), changing a skill (through education and training) or changing the technology used to carried out the practice (through the introduction of new technologies). This also applies to sustainable consumption practices [8,67]. Skills or meanings may be learnt or influenced through changes in context (e.g., visiting a local farmers market instead of a supermarket to purchase food).and technology can be modified to reduce the use of resources (e.g., low-flow showerheads instead of regular ones) [64]. It is in the realm of changing the meaning element that socio-psychological theories have an opportunity to share insights.

In the context of the home, changes in resource consumption have often been studied for individual practices [21,57,68–77]. However, individual practices form part of the HSOP and should not be targeted in isolation. The HSOP is a network of the practices performed by home occupants on a daily basis forming part of a routine [21]. Routines, composed of practices which are reproduced in a sequential manner (e.g., shower, breakfast, drive to work, etc.), overlap and interlock with one another [21,56], creating a home equilibrium (Figure 1a). These practices have specific temporal and spatial characteristics [78], which means that when performing practice in a specific location such as the home, occupants must consider other occupants' own practices and routines so as to coordinate them from a time and space perspective. For instance, the practice of showering is dependent on the shower being unoccupied at a particular moment in time.

Intervening in the HSOP to alter resource consumption can be done through changing an element of the practice as previously discussed. However, this requires all occupants to adopt the change. The introduction of new practices (e.g., checking an IHD) require established routines to become destabilised (Figure 1b) and realign, finding a new equilibrium (Figure 1c). Unless a realignment of everyday practices occur, new practices cannot become embedded in the HSOP and fulfil a specific role. Consequently, they might not be adopted by the home occupant.

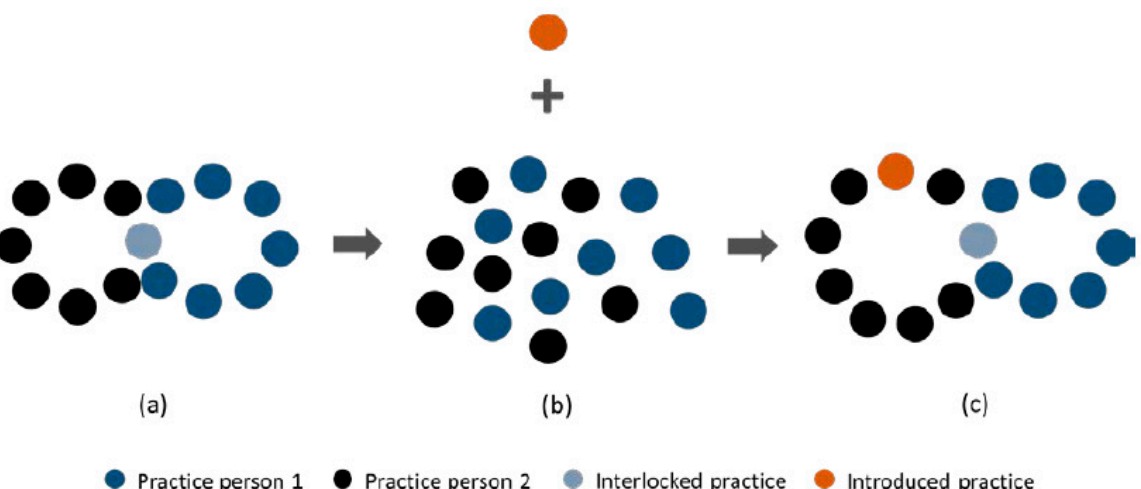

**Figure 1.** Equilibrium of practices in the home. (**a**) Original home equilibrium, (**b**) Destabilisation of the home equilibrium through the introduction of a new practice, (**c**) Realignment of practices and establishment of a new home equilibrium [56].

Encouraging resource reduction and efficiency can help to promote sustainable consumption, but this encouragement assumes that behaviours are static over time and are the result of conscious customer decisions and available technology [20]. It also ignores the often surprising links between seemingly unrelated practices [25]. SPT is useful for targeting inconspicuous practices and those that are deeply rooted in the normal living behaviour of individuals [25,79]. Practices are dependent upon the institutional arrangements, household context, economic influences and cultural traditions of the environment that the practice exists in [49]. Hence, the scale implications of practices and behaviours need to be considered when discussing their influence over resource consumption. The reproduction of practices and their proximity in space and/or time influences their degree of interlocking [21,61,67]. Consecutive practices that are part of a daily sequential personal routine are highly interlocked. For instance, morning weekday personal showers are highly interlocked with the practice of breakfast, which is in turn highly interlocked with the practice of transport to work. It is posited [21,61,64,67] that practices are highly interlocked when they occur at a similar time and at the same place (Figure 2). For instance, person 1 (P1) having a personal shower in bathroom 1 will influence person 2 (P2)'s shower in bathroom 1 if it is carried out at a similar time because P1 and P2's showers are limited by P1 and P2's own interlocked routines. Practices that occur at different times and in different places may not show the same degree of interlocking. Practices may be lightly or indirectly interlocked if they occur at the same time but in different places. For instance, the water heating technology may not work if P1 and P2 have personal showers at the same time but in two separate bathrooms, indirectly interlocking the two showering practices. Conversely, practices that occur at the same place but at different times may also indirectly affect each other.

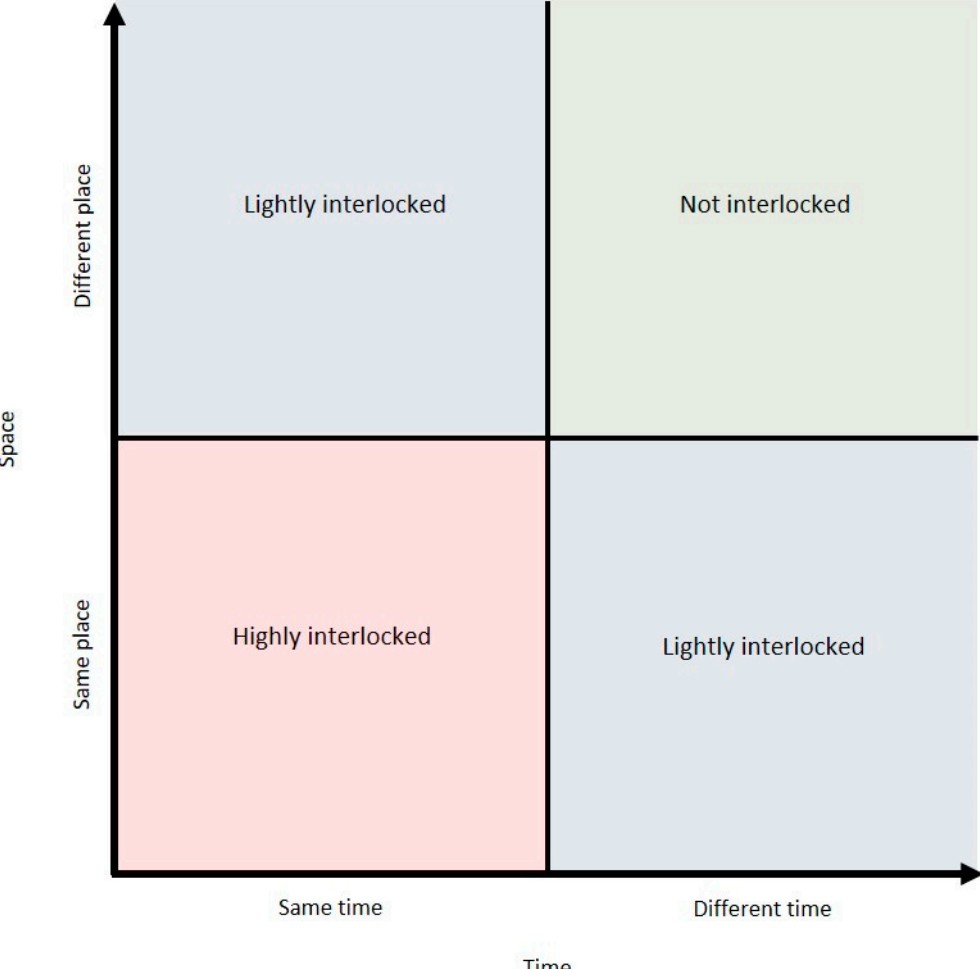

**Figure 2.** Interlocking degree of practices according to their reproduction in space and time.

### 3. Chalk and Cheese Debate

The merge of SPT with other theories has been previously attempted [80–83], these attempts focus on the individual and use SPT to deliver unique insights into daily practices. For instance, researchers have sought to merge theories of governmentality and SPT [84] to produce complementary insights bridging some of the weaknesses found in each of the theories. However, the latter approach has been either top-down or too focused on mundane daily practices to result in workable policy insights. Social practices have also been examined alongside transition theory, utilising the multi-level perspective (MLP) understanding to place practices at different socio-technical levels [85–87]. Building end users are examined through facility management and SPT in [88]. Some of the founders of contemporary SPT including Foulds and Macrorie utilise sociotechnical systems to understanding everyday practices [89,90]. However, the merge of socio-psychology theories and SPT has been a focus of disagreement termed the Chalk and Cheese debate.

Shove in her original (2010) article argues that psychological or behaviour change models have received too much engagement by policymakers and this has resulted in ineffective resource consumption policies being implemented [15]. The argument is made that psychological theories are confusing because while they posit that there are many factors influencing behaviours, they also claim that behaviours are deeply embedded in social situations, institutional contexts and cultural norms, locking consumers into particular habits. Shove outlines that the Attitude- Behaviour-Choice (ABC) model of policymaking (based in socio-psychology) has failed because of the value-action gap between individuals' reported environmental values and their behaviours. Shove argues that the ABC approach fails to take into consideration the practices of everyday life that affect the resources used by consumers [15]. According to her, behaviour change theories are based around the incorrect assumption that social change is to be dependent upon values and attitudes which drive the behaviour that individuals choose to adopt [17].

In rebuttal, Whitmarsh and colleagues argued that the dismissal of psychological models is ill-conceived [23]. They also note that other authors [91] refer to the C in the ABC as context rather than choice. Context considers the broader situation that the individual is situated in while choice focuses on the individual state of mind and actions. Considering context is vital in understanding social practices as it influences the elements of practice and how technology is used [41].

Whitmarsh and colleagues argue that if a pure SPT approach was to be incorporated into policy, individuals would run the risk of being excluded from societal decision making and participation, due to the focus of analysis moving to the practice itself, instead of the individual and their state of mind [23].

In response, Shove [22] argues that the two schools of thought have fundamentally different ontologies. The SPT ontology is that the world is populated by social practices and their interconnected elements, while socio-psychology theories are focused on individuals and their behaviours [16]. While we acknowledge this to be true, we do not discount the usefulness of both theories in providing insight into resource consumption patterns and practices of individuals, homes, communities and societies [92]. In fact, researchers are beginning to incorporate these theories into new models. Nevertheless, they require more empirical evidence to determine their success in reducing domestic resource consumption [31,93]. The theoretical differences between the two schools of thought are less relevant when considering interventions and policy recommendations [94].

The next section examines the different scales for application of these theories, whereby socio-psychological theories can offer useful insights for influencing the meaning of practices and SPT can offer insights on the material and technical structures that constitute practices [94].

### 4. Research on Space and Time Insights

Human choices are dependent upon the conditions under which the choice is made: these choices have temporal and spatial dimensions [17]. This was identified early on in the SPT literature by Schatzki, stating that practices are both anchored in and dispersed across space and time [19]. We posit

that insights from behaviour and practice theories have different roles based on the temporal and spatial dimensions they are being employed in. Practice insights can result in almost immediate changes in the way that practice is performed (for instance, if adopting a new technology), while behaviour insights often entail a longer-term cultural and societal shift (as it requires a shift in values and social norms) and can target the meaning element of practice.

The scale is an important consideration in the study of practices and behaviours. Practices depend upon individual performances for their continued survival and it is individuals who carry and integrate the necessary skills and knowledge that make each performance possible [25,46]. Practices are often nested in each other and have complex relations in the SOP. Practices are dynamic and mobile through their reproduction as performances and as such, exist in various temporal and spatial locations. This has also been termed time-space whereby space is simply a place to carry out particular activities or practices at a certain time [95]. By focusing on the actions of individuals rather than the individuals themselves, SPT brings aspects of space and time to the forefront of the decisions of everyday life [34]. SPTs have been critiqued for focusing more on the performances of practices rather than the mental or emotional state or events. Socio-psychology theories can fill this gap by directing attention to the values, norms and mind frame of individuals when they are engaging in the performance of an action [24,96].

In the SPT literature, practices are referred to in two different states, a practice-as-entity is a practice that endures over space and time and is composed of the three elements (i.e., meaning, technology and skill), while a practice-as-performance is when the elements are brought together and are different each time because of the context [97]. The practice-as-performances can be used to collect data on activities and resource consumption for analysis while practice-as-entities are used to sketch patterns of historical development and understand context [18]. Behaviour change insights may be suitable for influencing practice-as-entities, whereas the practices-as-performance may be more targeted through SPT insights such as changes to the technology or skills used to perform the practice.

Practice insights should also consider the context in which they are being performed, this harks back to the Chalk and Cheese debate and the role of the C in the ABC method: context or choice. Context is vital in understanding why a practice may be performed the way it is: people may choose to cycle to work because driving takes too long due to traffic congestion, not because of environmental concerns [94]. SPT provides this insight through considerations of meaning, while socio-psychology insights may miss this reasoning altogether if the context is not considered, although in some work it is [94]. Recent work has examined the material (or technology) element separately from the practice and discussed how the materials used in a practice influence space and time that the practice, and other practices, are performed [98]. Engaging in one practice rules out engagement in another at the same time and can then influence what practices are performed when and in what space [50]. It has been suggested by multiple researchers that to instigate change, focus needs to be placed on the junctions in space and time that constitute opportunities for change and innovation [67,70,78,99–101]. This may occur at times such as moving house, targeting new technology being installed or new communities that can influence social norms [56,61,64].

Research on the different spatial aspects of practices has included those of travel, snacking and recreation [51,102]. Due to changes in technology, the daily commute now involves many different practices that previously would have been performed at home or work. These include reading, writing, making phone calls, homework, self-care, drinking and eating [103]. Socio-psychologists have studied the habitual routine of travel choices and how to influence these through choice option, information and situational cues [104]. Those who have a strong habitual routine in their travel practices are less influenced by information and cues, however, habits can be broken through manipulating either accountability demands or the level of attention paid to the actions of the behaviour. The physical infrastructure used for mobility can influence what choices people make in how to perform practice. For instance, a high degree of car road networks and poor public transport options can influence the level of care practices performed [105].

The timing of the performance of behaviours and practices is a pertinent topic in relation to resource consumption, particularly in the home [101]. Recent work in the SPT literature has brought the temporal aspect of practices under analysis. This work concludes that time is socially constructed, with clear divisions between practices performed during the weekdays and weekends [78]. Some practices require a fixed location either in time or space to be performed, such as drying of washing can only occur after the clothes have been washed and this occurs with particular technology and skills [50]. The temporal placement of the actions and the associated network of relations that surround it, as investigated in socio-psychological theories, influences individuals actions [99].

Peak loads of resource consumption in households are associated with different forms of synchronizations, or particular behaviours and practices being performed together [50,106], as well as meanings and values associated with convenience and control in daily life [70]. For instance, the performance of energy consumption practices in households is tightly interlocked with the temporal patterns of the daily practices of the residents in the house [70,78]. Peak demand practices are performed during the evening, particularly between 17:00 to 21:00, when residents are preparing dinner, using electronic devices and keeping thermally comfortable with mechanical technology [107]. With the rise of renewable energy, the demand for energy during the peak hours of the morning and evening do not match when the energy is being generated [108]. Residents may be unwilling to alter their energy-intensive practices to match the time when energy is being generated even if it would change their resource carbon footprint (such as having shorter showers [61,64] or use less energy in the evening [41]). Recent studies have examined this further, finding that washing has the greatest time dependence in practices each week, being performed at a regular time by households, while using computers is the least time-dependent [78]. These findings can be used to develop interventions into resource consumption of both individuals and households.

## 5. The Home, Community and Societal Scale Applications

Individual behaviours and practices are often studied in-situ as this provides an easier contact point for the researcher. However, these actions do not occur in a bubble, devoid of contextual influence or never being performed in different locations [78]. Practices and behaviours tend to cluster together in particular places, being performed by many people in similar ways [78]. These can be based around the home scale, the community scale and the social scale. Other authors have referred to these scales as the individual, inter-personal networks, community, segments and population [109]. Inter-personal networks consisting of families, households or social groups. Community spaces are where people who share values or perform similar activities together clusters, such as at sports centres or religious gatherings. This can be linked to ideas of community of practice [25,110] where communities are the site of learning by individuals. Interventions are applicable for each level, taking advantage of social connections, temporal elements and domain influences over the behaviour [109].

To create long-lasting change in favour of sustainable consumption, we propose that all levels of society need to be targeted. The schematic in Figure 3 outlines how we propose SPT theory and behaviour change theories can best be applied at different scales. Practice insights can be used to achieve quick changes in a specific physical space (e.g., the home) while conserving established routines and respecting the HSOP where multiple individuals interact. This can be obtained through the implementation of technology that does not interfere with the timing or order of practices. Behaviour change, on the other hand, requires a shift of societal and cultural norms to be effective. Values and norms driving individual behaviours are usually present at all levels of society and are independent of space and time.

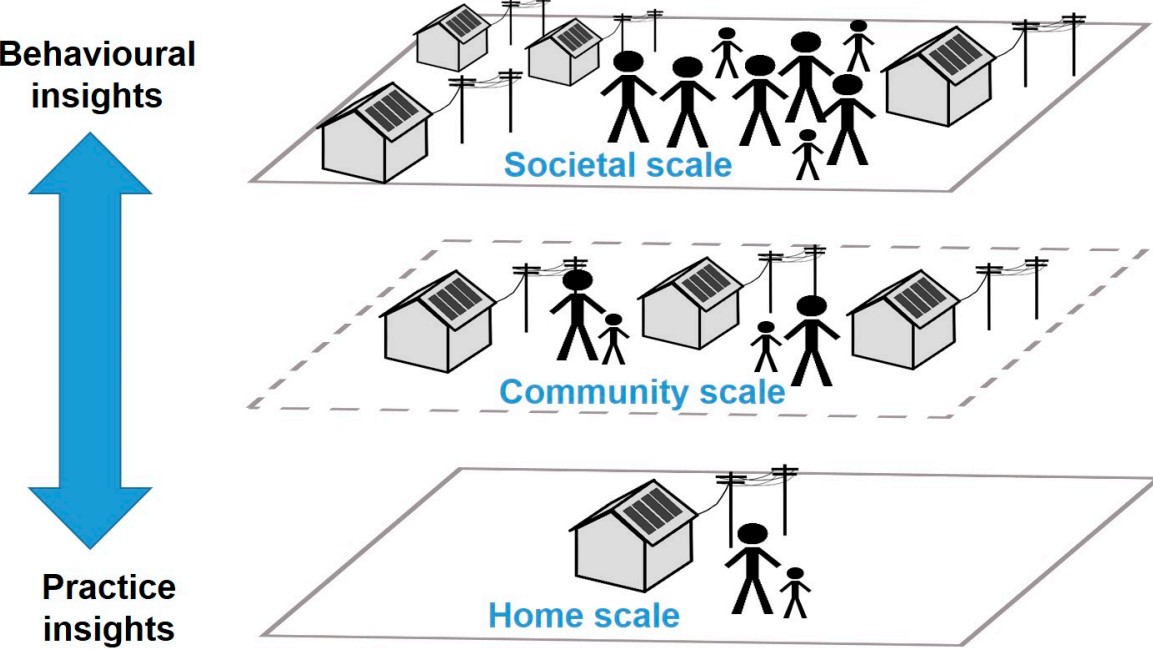

**Figure 3.** The different scales that behavioural and social practice theory (SPT) insights can be best targeted.

*5.1. Home Scale*

Interventions through socio-psychology methods in the home may only last for a short period of time, while design changes based on SPT insights will last for years, if not decades [111,112]. In spite of the limited data, the few existing longitudinal behaviour change studies suggest that some persuasive methods have longer-lasting effects than others in the space of the home [13]. Information campaigns, for example, might be effective for a short-term solution, such as saving water in summer [113]. IHDs can be a good tool while it is a novelty, but if it does not meet consumer needs, the technology quickly becomes idle and its effectiveness might only last a few months. Other interventions such as teamwork [114] and audits, which demand a higher level of personal engagement, might last a few years. Previous research has shown that behaviours end up reverting to what they originally were after the behaviour change intervention is interrupted, even if nudges or prompts are utilised [14].

Behaviours are influenced by close relationships as well as the wider society and culture and their insights would be better applied at a community and societal scale [15,65,115].

Design modifications at a household level based on SPT insights result in the change occurring to resource use [97]. For instance, changes can be made to the physical building system (e.g., the addition of such as insulation, shading devices or solar panels), resulting in a reduction in resource consumption. Technology can also be used to dis-interlock practices through automation, enabling lasting reductions in energy, water and resource use [21]. Automation is a once-off change, allowing a 'set and forget' mentality to drive the practice in future, reducing the need for human intervention and thus failure.

The replacement of current technology with technology that enables change through everyday practice can have a long-term effect on resource reduction. Rather than requesting that households turn off their standby appliances manually, for example, appliances without a standby mode could be provided instead [56]. The success of innovative technology is dependent on user knowledge and can be hindered by rebound effects. However, the implementation of practice-based design and co-creation is a contemporary approach which ensures that users not only take part in the design of a technology that is needed in the long-term, but also understand its function [116–119].

When technology is used to modify a practice (through SPT insights), it targets the specific space where the practice is carried out, without modifying the HSOP or the timing of the practice. This can enable change without requiring persuasion [14].

*5.2. Community Scale*

Community spaces are where people who share similar values or perform similar activities together [109]. Insights from both SPT and socio-psychological theories can be applicable at this scale, particularly regarding social norms, shared practices and information diffusion. Values and practices are co-constructive, where a person's beliefs, ethics and worldviews simultaneously shape and are shaped by their performance of practices [120]. Practices and behaviours that these insights have been applied to previously include the practices of mobility or food shopping [105,119,121,122]. Applications from the temporal studies of practices and behaviour can be applied at this scale through targeting when people are utilising the space if there are peak times that they are there and using resources that would be more effective to target them than others.

The ideas of both socio-psychological theories and shared practices have been discussed in the literature as a "community of practice" [25,110] where communities are the site of learning by individuals. Interventions in practices and behaviours that consider communities of practice have been shown to be effective in research targeting changes in driving cars within small villages, cooking for large groups of people once a week, shared community gardens and shared compost systems [123,124]. Through creating a sense of community between people, the impact that practice or behaviour has through sharing skills and being visible is increased, normalising what is being done and highlighting the actions of a group and the need to change social practices for sustainable impact [101]. These insights can be effective in changing practices and community perceptions of sustainable consumption [64,125,126].

Changing the skill of practices performed is also a way to target interventions on a community scale and can be achieved formally through education or social learning, such as through demonstration workshops or discussions amongst friends [69]. These groups can be communities of practice and be used as a targeted place to initiate practice and behaviour changes. Changes in skills, however, can be assumed to take longer to generate desired outcomes, as it is necessary for the skill to be learned and implemented [127].

*5.3. Societal Scale*

As previously discussed, the meaning relating to practice is closely connected to the context of where the practice is carried out and of what is socially acceptable. For instance, if the use of an electric heater is the norm to achieve warmth in the home, then the use of blankets, for example, is not considered a viable solution. This has been discussed in previous research, where residents who would usually not use auxiliary heating to keep themselves warm at home, will use it when friends or family visit because they believe it is the social norm [61].

Modifying the meaning of practices requires a societal shift, which is where socio-psychology theories can be used to best effect [64]. Social and structural changes [99], if perpetrated by government and organisations, or communities of practitioners, can affect cultural and social norms, along with the diffusion of knowledge [99]. Research has found that individuals perceive the solutions to climate change to be at the national or organisation level [99]. This highlights the expectation that individuals have that these changes will be disseminated from a societal scale. This can be through policy changes, information and education campaigns, and long term changes to cultural and social norms that may take generations to take effect. An example of this is the importance of change agents in the energy sector. These individuals can champion certain ways to perform practices that use fewer resources or shift the timing of the performance of energy-intensive practices through education or social norm campaigns [128].

Socio-psychology behaviour change programs have previously been applied at a societal scale to effectively encourage the use of seatbelt in cars. Results from multiple decades show that effective and well-planned media and enforcement campaigns can have a positive impact on seat belt usage rates [129,130]. Other research shows how an SPT understanding can be applied to enable change in specific places. Through providing people with belt-positioning booster seats (technology), education on its use and benefits (skills) and a social marketing campaign to emphasise social norms (meaning), effective behaviour change was achieved [131]. These examples show how on a societal scale, both socio-psychological and SPT insights can lead to enabling people to change effectively.

## 6. Conclusions

The significant challenges posed by climate change and overpopulation require an understanding of resource consumption across all scales of society. Policy implications should consider the reach and durability of various interventions targeting the reduction of resource use (e.g., energy, water, materials) if they are to be applied effectively [68]. Persuasion and enabling methods are usually not aligned in behaviour change studies as they are seen as two distinct methodologies. We argue that they are in fact trying to achieve the same objective, which is to reduce resource consumption in households but approaching it through different angles. On the one hand, socio-psychology uses persuasive methods to address personal energy use from a top-down approach, delivering social norms and providing information. On the other hand, SPT's enabling methods are very focused on the elements of a practice that are utilized in the performance and addresses consumption from a bottom-up approach. Rather than being conflicting methods, the two schools of thought complement each other, filling each other's gaps, informing and modifying attitudes while enabling long-term changes that bring value to the user.

The timing of the action is key because the practice or behaviour is limited by the context and influenced by other concepts such as comfort and convenience. Given the complexity of the home system, this paper suggests that rather than disestablishing, realigning and recreating interlocked connections and practices, automated technologies and design changes could enable improved resource efficiency. Automated practices can be performed at flexible times and in concert with other household technologies, removing the need for the occupants to be directly involved in the performance of the practice. Yet, they are still required to meet occupant needs and skills to be able to work effectively.

Socio-psychology theories have traditionally informed resource reduction in households from a user point of view. Their aim has been to modify behaviour by changing existing attitudes, knowledge and values without necessarily understanding the reasons behind existing behaviours, but rather focusing on the use of persuasion to promote change. It is imperative that consumers understand the implications of their actions and adopt a positive attitude towards resource consumption, but changing behaviour may involve perceived lifestyle impacts or change of embodied habits that require high levels of commitment which may, or may not, be feasible in everyday life. Studies testing the long-term effects of socio-psychology interventions suggest that the effects of most persuasive behaviour change methods last only between a few days and a few years, stopping as soon as making changes becomes too difficult, disruptive or loses the novelty effect. SPT offers a different approach to close some of the gaps found in persuasive interventions through investigating what drives people's practices and the reason for certain behaviours, that is, what and who is influencing the user in question. As highlighted by SPT researchers, people do not consume resources as such, but resources are used during the performance of a practice. Technologies used to achieve these practices are therefore a key part of the practice and have a direct impact on resource use. These changes can be effectively applied at the scale of the home to drive resource change. Insights from socio-psychological theories can be best applied at the community and society scale where changes to values and social norms, along with information diffusion and education can drive long-term change. Understandings of communities of practice and change agents from the SPT literature can assist in delivering interventions.

Having an understanding of what drives practices and behaviours at the individual, home, community and societal scale allows the focus of policymakers to be on designing and implementing

effective interventions that will enable sustainable consumption in society. It also enables designers of new technologies to focus on the material elements and skills that can assist users to make permanent changes rather than just convince them that change is necessary.

**Author Contributions:** Conceptualization, J.K.B., C.E. and G.M.M.; methodology, J.K.B. and C.E.; validation, J.K.B. and C.E.; formal analysis, J.K.B. and C.E.; investigation, J.K.B. and C.E.; resources, J.K.B. and C.E.; data curation, J.K.B. and C.E.; writing—original draft preparation, J.K.B. and C.E.; writing—review and editing, J.K.B., C.E. and G.M.M.; visualization, J.K.B. and C.E.; supervision, G.M.M.; project administration, J.K.B., C.E. and G.M.M.; funding acquisition, G.M.M.

**Funding:** This research is funded by the CRC for Low Carbon Living Ltd (Project number NP2006) supported by the Cooperative Research Centre's program, an Australian Government initiative.

**Conflicts of Interest:** The authors declare no conflict of interest

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
