# Peer review of "Understanding Resource Consumption in the Home, Community and Society through Behaviour and Social Practice Theories"

_sustainability, doi:10.3390/su11226513_

Round 1

Reviewer 1 Report

It is not an easy reading, mainly of the first page. It is due to detailed information. I will suggest more figures and exemples about the concept. For instance, the diferences between, attitudes, beliefs and values, with na application.

It is difficultt to read, it is an "heavy text", and somestimes it is needed to reread for understanding the meaning of the both  theories and .Do not forget, this paper is a scientific paper to be read all around the world.

Although this suggestion, it is a very good paper in my oppinion. However, it is missing an experiment to validate your analysis. It would  be the next step, I suppose.

Author Response

Comments

Response

I will suggest more figures and exemples about the concept. For instance, the diferences between, attitudes, beliefs and values, with na application.

This has been addressed in the second paragraph of section 2.1 already.

It is not an easy reading, mainly of the first page. It is due to detailed information. It is difficultt to read, it is an "heavy text", and somestimes it is needed to reread for understanding the meaning of the both  theories and . Do not forget, this paper is a scientific paper to be read all around the world.

The paper has been revised in sections to account for this

Although this suggestion, it is a very good paper in my oppinion. However, it is missing an experiment to validate your analysis. It would  be the next step, I suppose.

The applications of the theories to a case study is outside the scope of this paper. Other papers published by these authors and others that feature case studies related to this are referred to throughout the paper.

Reviewer 2 Report

The article discusses based on relevant literature the potentially complementary role of socio-psychological and social practice theories. The proposed model of "Home-Community-Societal level" is interesting. However, it could be argued, that SPT plays a role especially at Societal level as well (large-scale infrastructures, political regulation of technologies, services etc.). Moreover, the discussion apperas to be somewhat a-political. The key role of political ideologies and power relations in shaping SPT as well as socio-psychological dimensions could have been adressed more explicitly. The mentioning of "change agents" as driver for bottom-up changes in social practices at the household and community level underestimates the significant role of STP-related political activities (for example, public transport, electric mobility, spatial planning, requirements for eco-design of products, buildings etc.). It might be interesting to enrich the article by including this important aspect in the context of socio-psychological and SPT approaches. Finally, the more recent socio-psychological focus on "nudging" has not been included at all. Why?

Author Response

Comment

Response

The article discusses based on relevant literature the potentially complementary role of socio-psychological and social practice theories. The proposed model of "Home-Community-Societal level" is interesting. However, it could be argued, that SPT plays a role especially at Societal level as well (large-scale infrastructures, political regulation of technologies, services etc.). Moreover, the discussion appears to be somewhat a-political. The key role of political ideologies and power relations in shaping SPT as well as socio-psychological dimensions could have been addressed more explicitly. The mentioning of "change agents" as driver for bottom-up changes in social practices at the household and community level underestimates the significant role of STP-related political activities (for example, public transport, electric mobility, spatial planning, requirements for eco-design of products, buildings etc.). It might be interesting to enrich the article by including this important aspect in the context of socio-psychological and SPT approaches.

There has not been much research undertaken on the political aspect of SPT and so it has been decided to be left out of this paper due to the already large coverage of factors of the two theories covered already and the specific focus on the applications to practices of individuals.

Finally, the more recent socio-psychological focus on "nudging" has not been included at all. Why?

This concept has been included, it was just not referred to as nudging. That term has been included in the sections now.